# Levocetirizine-Loaded Electrospun Fibers from Water-Soluble Polymers: Encapsulation and Drug Release

**DOI:** 10.3390/molecules28104188

**Published:** 2023-05-19

**Authors:** Lan Yi, Lu Cui, Linrui Cheng, János Móczó, Béla Pukánszky

**Affiliations:** 1Laboratory of Plastics and Rubber Technology, Department of Physical Chemistry and Materials Science, Budapest University of Technology and Economics, H-1521 Budapest, Hungary; 2Institute of Materials and Environmental Chemistry, Research Centre for Natural Sciences, ELKH Eötvös Loránd Research Network, H-1519 Budapest, Hungary

**Keywords:** levocetirizine, water-soluble polymers, drug distribution, pH, drug release, modeling

## Abstract

Electrospun fibers containing levocetirizine, a BCS III drug, were prepared from three water-soluble polymers, hydroxypropyl methylcellulose (HPMC), polyvinylpyrrolidone (PVP) and polyvinyl alcohol (PVA). Fiber-spinning technology was optimized for each polymer separately. The polymers contained 10 wt% of the active component. An amorphous drug was homogeneously distributed within the fibers. The solubility of the drug in the polymers used was limited, with a maximum of 2.0 wt%, but it was very large in most of the solvents used for fiber spinning and in the dissolution media. The thickness of the fibers was uniform and the presence of the drug basically did not influence it at all. The fiber diameters were in the same range, although somewhat thinner fibers could be prepared from PVA than from the other two polymers. The results showed that the drug was amorphous in the fibers. Most of the drug was located within the fibers, probably as a separate phase; the encapsulation efficiency proved to be 80–90%. The kinetics of the drug release were evaluated quantitatively by the Noyes–Whitney model. The released drug was approximately the same for all the polymers under all conditions (pH), and it changed somewhere between 80 and 100%. The release rate depended both on the type of polymer and pH and varied between 0.1 and 0.9 min^−1^. Consequently, the selection of the carrier polymer allowed for the adjustment of the release rate according to the requirements, thus justifying the use of electrospun fibers as carrier materials for levocetirizine.

## 1. Introduction

One of the most convenient methods for the application of a drug is oral administration [1,2,3]. This approach has many advantages, including the cooperation of patients, formulation freedom and cost effectiveness. Consequently, pharmaceutical companies prefer the preparation of drug formulations which are administrated orally. However, in many cases, the physical and chemical characteristics of drugs do not favor oral administration because of limited solubility, poor permeability or other reasons [4,5,6,7,8,9], such as bad taste. The most important drawback of oral administration is low bioavailability; thus, various measures must be taken to improve it. Accordingly, a large number of approaches have been applied to develop formulations offering easy administration and high bioavailability at the same time.

Various physical and chemical approaches have been developed to improve bioavailability. Chemical methods, i.e., the modification of the chemical structure of a drug [10,11], are usually complicated and expensive and are not always successful; thus, physical methods are preferred. The most important of these are the decrease in particle size [3], the change in crystallization conditions [12], the preparation of solid dispersions [13,14,15] and solubilization [16,17]. The simplest of these approaches is the decrease in the size of drug particles by grinding or micronization [18,19]. Active molecules are often solubilized by complexation and by the combination of the drug with a host molecule, most frequently with cyclodextrins or surfactants [17,20,21]. 

A wide variety of devices have been prepared for the controlled delivery and increased release rate of the most diverse drugs and active components from nanoparticles [22,23] to membranes [24] to nanofibers [25,26,27,28]. These devices were fabricated from a wide range of materials including natural [28] and synthetic polymers [26], hydrogels [22,27,28] and composite materials [23,24,26]. Nanofibers, and specifically electrospun fibers, are often the preferred solution to prepare drug formulations with improved bioavailability. This method is used more and more frequently in medicine, because it has considerable advantages [29]. Spinning technology offers flexibility in the production of fibers and in formulation and frequently offers the possibility of controlling release kinetics or generating other medical devices, e.g., biotextiles or nanofibrous scaffolds for tissue engineering [30,31]. Fiber characteristics depend on a number of factors including the characteristics of the components, i.e., the polymer, the solvent and the drug, as well as on processing parameters, such as voltage, the pump rate and the distance to the collector [32]. The release of the drug from the fiber formulation is influenced by all of these parameters as well as by the interaction of the components, which may change the form and location of the drug in the fibers and thus the release kinetics [33,34]. Although many parameters influence drug release, they also offer a possibility of controlling solubility and the dissolution rate at the same time.

Levocetirizine dihydrochloride (Levo) is a third-generation non-sedative antihistamine developed from a second-generation drug, cetirizine. According to the Biopharmaceutics Classification System, Levo is a BSC III drug; thus, its solubility is excellent, but its permeation ability is poor. In the past few years, several attempts have been made to formulate Levo to improve its solubility and bioavailability or simply to mask its bitter taste. Similarly to other drugs, one of the approaches used was its complexation with β-cyclodextrin to prepare a fast-dissolving drug delivery system [35]. Besides resulting in fast delivery, the formulation also masked the bitter taste of the drug. Levo has also been encapsulated into chitosan nanoparticles [36], leading to a prolonged release. Another approach applied quite often is the preparation of thin films containing the drug which is claimed to result in improved bioavailability [37,38]. Such films are prepared from water-soluble polymers such as HPMC and PVA, resulting in a faster drug release [39,40]. Jahangir et al. [41] studied the interaction of Levo with various water-soluble polymers and found that strong interactions do not develop between the drug and the polymer; thus, the carrier materials do not influence the effect and efficiency of Levo. According to our best knowledge, no attempt has been made to incorporate Levo into formulations consisting of electrospun fibers prepared from water-soluble polymers.

The results published in the available literature indicate that the studies undertaken with levocetirizine up to now have had various goals. Some wanted to increase the release rate, while others aimed at prolonged release or simply intended to mask the bitter taste of the drug. Considering these sometimes contradictory aims and results, the goal of our study was to prepare formulations from Levo by incorporating it into electrospun water-soluble polymer fibers. The polymers selected were HPMC, PVP and PVA, all approved and used in pharmaceutics. The water solubility of the polymers allows for rapid dissolution, thus modifying the release rate and bioavailability. The electrospinning conditions were optimized during the study and the morphology of the fibers was thoroughly characterized. The form, location and distribution of Levo in the fibers were studied by various methods, and then the drug release was determined as a function of the pH of the dissolution medium. The release rate was modeled using the Noyes–Whiney equation and the results were interpreted from the point of view of practical applications.

## 2. Results and Discussion

The results are discussed in several sections. The characteristics of the components having some bearing on the preparation of the device and the efficiency of drug release are considered first. Interactions and miscibility are analyzed next, followed by a presentation of fiber characteristics. The distribution of the drug in the device and its location are shown in the following section, while the results of the drug release study and the quantitative analysis are discussed in the last section, together with some notes on their relevance for practice.

### 2.1. Component Properties

In formulations containing components other than the drug, interactions may take place among the components, which influences the efficiency. These interactions depend on the chemical and physical characteristics of the components. Levocetirizine contains several rings and an acidic group, the latter being able to form strong intermolecular hydrogen bonding with some of the carrier polymers used in this study. Levo is a crystalline material as shown by the SEM micrograph of the powder (Figure 1A), DSC measurements and by the XRD pattern recorded on the latter (Figure 1B). According to literature sources, the melting point of the crystals is around 205–208 °C [42], but DSC measurements yielded a melting temperature of 222 °C on the specific sample used in this study. A high degree of crystallinity hinders the dissolution of the drug and influences its dispersion in any carrier material. Amorphous compounds, on the other hand, dissolve faster and their bioavailability is usually better.

As described in the introduction, three water-soluble polymers have been selected as carrier materials for this study. All are approved materials for pharmaceutical use. Their solubility in water allows for the preparation of formulations that dissolve fast, thus allowing for an efficient release of the drug. However, the distribution of the drug and its physical form, crystalline or amorphous, influence the efficiency of the device very much. All contain strongly polar groups capable of dipole-like interactions and of forming hydrogen bonds. PVA is a crystalline polymer, while the other two are amorphous. The physical structure of the polymers also influences their dissolution, and in a previous study, PVA was shown to dissolve slower than the other two polymers, thus determining the rate of release and its dependence on pH [43]. Although Jahangir et al. [41] found that strong interactions do not form between Levo and the three polymers used in this study, based on the chemical structure of the components, some doubts might be raised about that claim. Accordingly, interactions and solubility are discussed in the next section.

### 2.2. Interactions and Solubility

As mentioned above, the interaction of the components might have a strong impact on the efficiency of the prepared device. Small-molecular-weight polar compounds usually dissolve to a very small extent in polymers with a smaller polarity. Billingham et al. [44], for example, reported that the dissolution of various stabilizers in polypropylene and polyethylene cannot be more than 0.1 or 0.2 wt%. Although the polymer used in this study contains a number of polar groups, the solubility of the drug in them must be known. Drugs are often added to formulations with a complete disregard for interactions and solubility. Similarly, the solubility of an active component in a solvent or a solvent mixture might be limited, just as in the case of BCS II drugs [34,43]. 

The easiest way to estimate interactions is the calculation of their Hildebrand solubility parameter by the group contribution approach. The solubility parameters estimated by Fedor’s method [45] for the main components used in the technology are listed in Table 1. One can see that the solubility parameters cover a wide range from about 20 MPa^1/2^ to the large value of water (47.9 MPa^1/2^). Hydrogen bonding interactions play an important role in this large number, but PVA and HPMC are also able to form such interactions, as mentioned above. The solubility of the drug in the polymers as well as in the solvents and solutions used in the fiber-spinning process might also cover a wide range. 

The solubility of the drug in the polymers was determined by a simple but efficient method used successfully earlier for the determination of the solubility of small-molecular-weight compounds in various polymers [34,46]. The method is demonstrated in Figure 2. The intensity of the characteristic absorbance of the drug is proportional to its concentration in thin films until the compound is dissolved and distributed at the molecular level. When the solubility limit is reached, the drug precipitates and forms a separate phase. Accordingly, the intensity of absorption does not increase further or just proportionally to the surface of the particles dispersed in the polymer. As Figure 2 shows, the solubility of Levo in HPMC is limited, much smaller than the amount added to the polymer during electrospinning. The discrepancy must be accounted for later. The solubility of the three polymers is listed in the third column of Table 1.

The solubility of the drug in the solvents used during the fiber-spinning process is considerably larger because of the entropy term of the free enthalpy of mixing (see Table 1). The solubility of the solvents was determined by preparing an oversaturated solution, separating the undissolved drug and determining the Levo content of the saturated solution by UV–Vis spectrophotometry. A separate calibration curve was constructed for all the solvents and buffers. The solubility of the solvents is larger than that of the polymers, but relatively small in DCM because of the lack of specific interactions. The values obtained for the buffers with different pH values are similar; the relatively small differences might result from the presence of ionic moieties in the buffers as well as from experimental error considering the simplicity of the method used. Nevertheless, it is clear that electrospinning starts from a homogeneous solution; however, phase separation occurs at the end of the process, and the drug forms a separate phase in the device.

### 2.3. Fiber Characteristics and Morphology

Electrospinning is a simple technology, but the productivity and quality of the fibers are influenced by a considerable number of factors and parameters. Accordingly, the technology must be optimized in order to obtain fibers of good quality. Our optimization started from the parameter set determined in a previous study [43], but it had to be modified somewhat because of the use of a new apparatus and the effect of the different drugs on the technology. The most important parameters optimized were the polymer concentration of the solution as well as the voltage, feeding rate and collector distance. The parameters obtained and used during the production of the fibers are collected in Table 2. In order to obtain good-quality fibers from PVA, 30 vol% ethanol was added to the spinning solution. The parameters listed in the table clearly show that an individual parameter set had to be developed for each carrier polymer.

The fiber-spinning process determines the quality of the fibers and the encapsulation of the drug into them. The specific surface area of the fibers depends on their thickness, which thus influences the rate of release. The morphology of the fibers obtained from the three polymers is presented in Figure 3. Very-good-quality fibers were obtained from all the polymers, although the shape and diameter of the HPMC fibers showed a larger variation than the same characteristics of the fibers prepared from the other two polymers. Because of the importance of fiber size, the diameter of the fibers was determined by digital optical microscopy (DOM). The distribution of fiber diameter is presented in Figure 4 as an example for fibers prepared from PVP. The figure clearly shows that the thickness of the fibers is uniform, and the presence of the drug basically does not influence it at all. The fiber diameters are in the same range, although somewhat thinner fibers could be prepared from PVA than from the other two polymers. The distribution of fiber diameters is narrow for PVP and PVA, but rather wide for HPMC, in accordance with the conclusion drawn from visually observing the SEM micrographs. The drug does not influence the diameter of the fibers considerably; i.e., it does not interfere with the fiber-spinning process. The fiber diameters and the width of the distributions are collected in Table 3.

Besides the morphology of the fibers, the structure of the components, both of the drug and the polymer, influences the rate of drug release. The crystallinity of the drug slows down its dissolution and thus release, while that of the polymer determines the location of the drug, because the latter can be located only in the amorphous phase of the polymer. The structure of the fibers containing the drug was studied by DSC and XRD measurements and is demonstrated in the example of the PVA fibers in Figure 5. According to the DSC measurements, the polymer is crystalline, but the originally crystalline drug (see Figure 1) is completely amorphous in the fibers. The same information is obtained from the results of the XRD measurements (Figure 5); i.e., the polymer is crystalline, but the drug is amorphous, which facilitates release and increases bioavailability. Both the polymer and the drug are completely amorphous in the case of the other two polymers, i.e., PVP and HPMC.

### 2.4. Distribution, Location

In most cases, the amount of an active component in electrospun fibers is assumed to be the same as in the spinning solution. Unfortunately, this is rarely the case, as the composition changes continuously during fiber spinning and subsequent drying, as shown earlier [34]. The presence and the subsequent evaporation of the solvent may result in phase separation and occasionally in the crystallization of the carrier polymer. This latter process leads to the further separation of phases, since the active component is excluded from the crystals. In our case, the initial drug content of the spinning solution was 10 wt% relative to the amount of the polymer, but the solubility of the drug in the polymers is much less: 0.7–2.0 wt%. Accordingly, the distribution, location and form of the drug are important issues.

The results presented in the previous sections showed that the drug is amorphous in the fibers. However, they can be located within or among the fibers. The presence of the drug in the fibers was checked by FTIR spectroscopy, and the result is shown in Figure 6 for the HPMC fibers as an example. The spectra of the drug, the polymer and the fibers are compared in the figure. The vibration of the carboxyl group at 1742 cm^−1^ clearly shows the presence of the drug in the fiber. The slight shifting and broadening of the band indicate that, contrary to the claim of Jahangir et al. [41], the polymer and the drug interact with each other, which is not very surprising. Similar spectra were obtained for the other two polymers as well. The amount of the drug encapsulated within the fibers after the spinning process was determined by dissolution and UV–Vis spectroscopy. The results showed that 88.6, 87.4 and 79.3% of the drug added originally are located within HPMC, PVA and PVP, respectively. The encapsulation efficiency is very good, especially compared to the result obtained earlier on the PLA fibers containing amoxicillin as the active component [34].

The drug is definitely incorporated into the device prepared by electrospinning, but it can be located within or among the fibers [34]. The crystals of the precipitated drug could be observed clearly among the fibers earlier on SEM micrographs [34]. In this study, no crystals could be found among the fibers in any of the polymers, but according to some micrographs, particles of the drug seem to be located within the fibers as a separate phase. Such particles are shown by white circles in the micrograph presented in Figure 7. Naturally, SEM micrographs cannot offer unassailable proof of the presence of the particles, but the results are in accordance with previous ones related to solubility (Section 2.2) and structure (Section 2.3).

### 2.5. Release and Dissolution

The goal of any drug formulation is to achieve the best bioavailability possible, which depends on solubility and permeation. The release of the drug was studied as a function of pH. Contrary to a previously studied drug, valsartan [43], the dissolution of the neat drug depended only slightly on the pH of the medium as shown by Figure 8. The dependence of the dissolved amount is inconsequent, which might be the result of experimental error, but always reaches almost 80%. The results show that, as a BCS III drug, the solubility of Levo is good. The situation is very similar in the case of the fibers. The release from the HPMC fibers is presented in Figure 9. The amount of the released or dissolved drug is very similar as for neat Levo, but differences can be seen in the rate of release. The release rate is the slowest at a pH of 1.2 and somewhat faster in distilled water and at a pH of 6.8. The effect of the type of the polymer on the rate of drug release is compared in Figure 10 at a pH of 1.2. The results clearly show that compared to the dissolution of the neat drug, the release rate is faster in PVP and slower in HPMC, while approximately the same in PVA as for neat Levo. Obviously, the rate of drug release can be adjusted by the selection of the carrier polymer.

The results presented in Figure 8, Figure 9 and Figure 10 indicate that the amount of the released drug is approximately the same for all the polymers under all conditions (pH), and it changes somewhere between 80 and 100%. The release rate seems to vary more significantly as a function of pH and the type of the polymer, but our visual observation does not allow for an exact estimation of the differences. A quantitative evaluation needs a model that allows for the determination of the rate of release. The kinetics of dissolution can be described by the Noyes–Whitney equation [46] as follows:(1)dctdt=A DV hcs−ct
where *c* is the concentration of the drug, *t* is the time, *A* is the surface area, *D* is the diffusion coefficient, *V* is the volume of the dissolution medium, *h* is the thickness of the diffusion layer and *c_s_* is the solubility. The integration of the equation and a rearrangement lead to the following:(2)$$c\left(t\right)=c_{s}\left(1-Exp(-kt)\right)$$
where *k* contains the constants of Equation (1), i.e., *k* = *AD*/*Vh*, and corresponds to the overall rate of dissolution. Equation (2) was fitted to all dissolution and release correlations to determine the solubility (*c_s_*) and the rate of dissolution (*k*).

Since the amount of released drug, i.e., the solubility, is practically the same in each case, we refrain from its presentation, but plot the overall rate of release as a function of pH in Figure 11. The value of *k* decreases for Levo with pH and also for the HPMC fibers but at a somewhat lower level. The release rates are similar for PVA, but increase with pH. The strongest dependence is shown by the PVP fibers; the rate constant of release increases from about 0.4 to around 0.9 min^−1^ at the highest pH studied. Unlike for valsartan, practically the same rates are measured in water (indicated by the filled symbols) as at a pH of 6.8; i.e., ionic strength does not influence release very much. The only exception is PVA, but the single, deviating point measured in water for this polymer might easily be the result of experimental error and needs checking in the future. Modeling confirmed the conclusions drawn from the visual observation of the dissolution and release isotherms and shows that the release rate depends both on the type of polymer and on pH. These dependences allow for the adjustment of release by the selection of the polymer according to the tract in which the drug is released.

## 3. Experimental

### 3.1. Materials

The Levo used in the experiments was supplied by Egis Pharmaceutical PLC (Budapest, Hungary). Three water-soluble polymers were selected for the study. HPMC (Methocel E5, HPMC, M_w_ 28,700 g/mol) was purchased from Colorcon Limited (Harleysville, PA, USA), PVP was obtained from Alfa Aesar (Tewksbury, MA, USA, M_w_ 58,000 g/mol) and PVA (Mowiflex LP, PVA, M_w_ 32,000 g/mol, hydrolysis rate 87–89%) from Kuraray (Tokyo, Japan). The chemicals used for the preparation of solutions and buffers, i.e., disodium hydrogenphosphate-2-hydrate, anhydrous sodium dihydrogen phosphate, ethanol, methylene chloride, sodium hydroxide and hydrochloric acid (37%), were purchased from Molar Chemicals, Hungary, while acetic acid was purchased from Merck (Darmstadt, Germany) and sodium acetate-3-hydrate from Reanal (Budapest, Hungary). All materials and chemicals were used as received. 

### 3.2. Solutions

HPMC was dissolved in a 1:2 mass ratio mixture of ethanol and methylene chloride. Ethanol was used as solvent for the preparation of PVP solutions. The concentration of the polymer changed between 20 and 50 wt% during the optimization of the electrospinning technology. PVA was dissolved in distilled water to prepare solutions containing the polymer at 15 wt%. In order to improve the quality of the fibers and find optimum conditions, water was replaced with 10, 20, 30 and 40 vol% ethanol. HPMC and PVP solutions were prepared by continuous stirring overnight. PVA could be dissolved at 70–90 °C in 20–30 min by intensive stirring. The active component, Levo, was dissolved in all three solutions at 10 wt% relative to the amount of the polymer.

Release experiments were carried out in four media, in buffers with three different pH values and in distilled water. The solutions were prepared in a calibrated measuring vessel with a 1000 mL capacity. The composition of the buffer solution with a pH of 4.0 and 6.8 was taken from the literature [34]. The solution with pH of 1.2 was prepared with 5.26 mL of concentrated (37%) hydrochloric acid. The acetate buffer (pH 4.0) contained 4.7 mL of concentrated (100%) acetic acid, 2.45 g of C_2_H_3_O_2_Na·3H_2_O and 2 mL of 1 M NaOH solution. The phosphate buffer with pH of 6.8 was produced from 6.12 g of NaH_2_PO_4_, 8.72 g of Na_2_HPO_4_·2H_2_O and 2 mL of 1 M NaOH solution. The pH of the solutions was checked with the help of a Metrohm 827 pH apparatus (Metrohm Ltd., Herisau, Switzerland) and was adjusted by adding the necessary amount of 1 M NaOH solution.

### 3.3. Electrospinning

Fibers were fabricated using the Spinsplit (Spinsplit LLC, Budapest, Hungary) electrospinning machine. Concentration, voltage, pump rate and the distance to the collector plate all depended on the characteristics of the solutions, i.e., on the combination of the polymer and solvent. The optimized parameters were different for the three polymers used. The time of fiber spinning changed between 5 and 60 min depending on the amount of fiber needed. 

### 3.4. Release Experiments

Release experiments were carried out on 10 mg fiber containing 1 mg Levo. All measurements were performed in triplicates. The fibers were placed into 100 mL solution, and 2 mL samples were taken intermittently after 1, 3, 5, 8, 10, 12, 15, 20, 25 and 30 min. The concentration of the drug in the samples was determined by UV–Vis spectrophotometry after calibration. Separate calibration curves were constructed for each of the four release media. After the determination of UV absorbance, the samples were replaced with the beaker containing the fibers. The experiments were carried out at room temperature without stirring. 

### 3.5. Characterization

The encapsulation of the drug into the fibers was checked by Fourier transform infrared spectroscopy (FTIR). Spectra were recorded using a Bruker Tensor 27A (Bruker Corp., Billerica, MA, USA) apparatus in the wavelength range of 4000–400 cm^−1^ at 2 cm^−1^ resolution with 64 scans. Levo and fibers cut into small pieces (2 mg) were mixed with KBR to prepare pastilles for the recording of the spectra. The spectra were evaluated using the Opus 2015 software. Encapsulation was also checked by dissolving 1.5 mg fibers containing the drug in 2 mL distilled water. The fibers were placed into capped glass vials and dissolved with stirring for 30 min to ensure complete dissolution. The amount of dissolved Levo was determined by UV–Vis spectrophotometry using a Unicam UV 500 spectrophotometer (Unicam Ltd., Cambridge, UK) after calibration. The measurement was performed in the wavelength range of 200–300 nm with 1 nm resolution and a scan rate of 240 mm/min. Experiments were run in triplicates. The solubility of the drug in the three polymers used as matrix was also determined by separate experiments. Solvent cast films were prepared from the polymers containing various amounts of drug, and the characteristic absorbance of the drug was determined by UV–Vis spectrophotometry. Solubility was determined from the break in the absorbance vs. concentration correlation upon the drug forming a separate phase.

The crystalline structure of the components was studied by differential scanning calorimetry (DSC) and X-ray diffraction measurements (XRD). DSC measurements were performed on 3–5 mg fibers or powder samples of the drug using a Perkin Elmer DSC IC apparatus (Perkin Elmer Inc., Waltham, MA, USA); samples were heated from 30–250 °C at a heating rate of 10 °C/min under N_2_ purge, cooled down at the same rate and then heated again. XRD patterns were recorded using a Philips PW 1830 diffractometer (Philips N.V., Amsterdam, The Netherlands). Measurements were carried out in the range of 2θ angles of 5–40° with 0.04° increments and 1 s/step rate at the accelerating voltage of 40 kV and exciting current of 35 mA. 

The morphology of the fibers was studied by digital optical (DOM) and scanning electron microscopy (SEM). DOM micrographs were recorded using a Keyence VHX 5000 microscope (Keyence Corporation, Osaka, Japan), and they were used for the optimization of the electrospinning process as well as for the determination of fiber thickness. SEM micrographs were taken by using a JEOL JSM 6380LA (Jeol, Tokyo, Japan) scanning electron microscope at the accelerating voltage of 15 kV.

## 4. Conclusions

This study on the encapsulation of levocetirizine, a BCS III drug, into electrospun water-soluble polymers indicated that this inherently crystalline drug is distributed in an amorphous form in the fibers. The solubility of the drug in the polymers used is limited, with a maximum of 2.0 wt%, but it is very large in most of the solvents and solvent mixtures used for fiber spinning. After the optimization of the fiber-spinning technology, good-quality fibers could be prepared from all three polymers. The size distribution of HPMC fibers is large, while the fibers prepared from the other two polymers have quite a uniform diameter. Most of the drug is located within the fibers, probably as a separate phase; the encapsulation efficiency proved to be 80–90%. The same amount of drug is released into the surrounding media almost independently of the type of polymer and pH. The rate of release, however, changed with a much wider range; the rate constant of release varied between 0.1 and 0.9 min^−1^. Consequently, the selection of the carrier polymer allows for the adjustment of the release rate according to the requirements, thus justifying the use of electrospun fibers as carrier materials for levocetirizine.

## Figures and Tables

**Figure 1 molecules-28-04188-f001:**
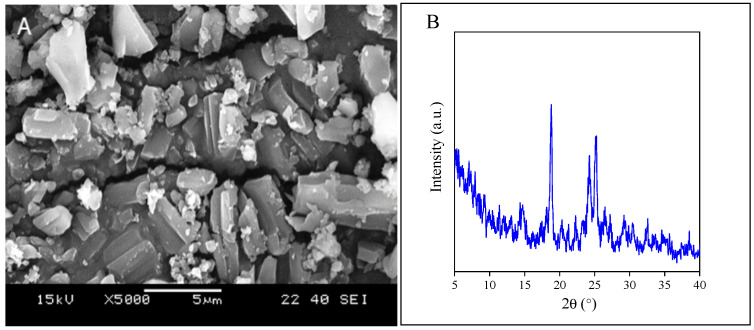
The morphology of Levo: (**A**) SEM micrographs recorded on crystalline particles; (**B**) XRD pattern of the drug.

**Figure 2 molecules-28-04188-f002:**
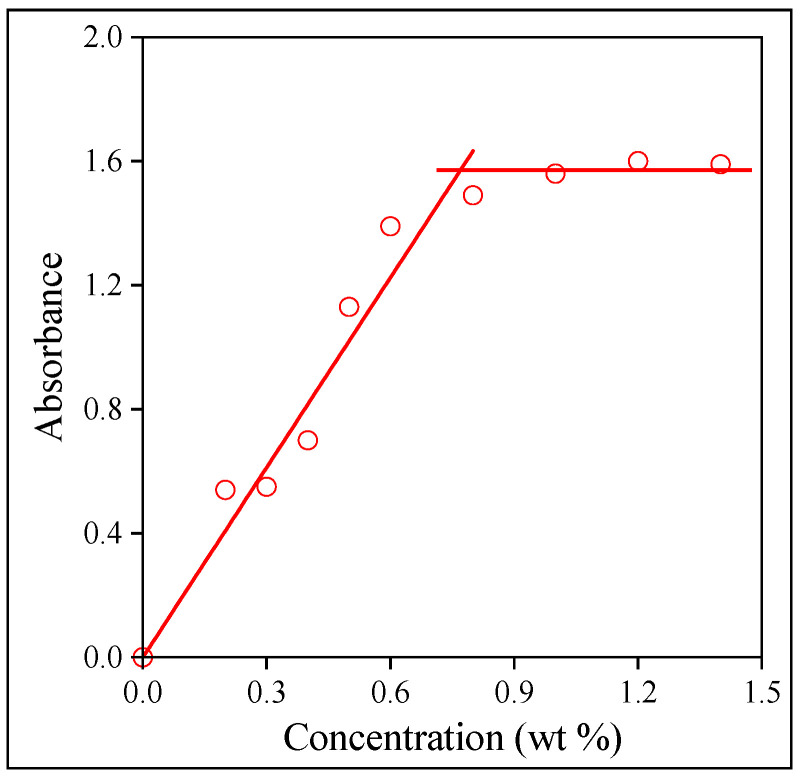
Determination of the solubility of Levo in HPMC films with UV–Vis spectrophotometry.

**Figure 3 molecules-28-04188-f003:**
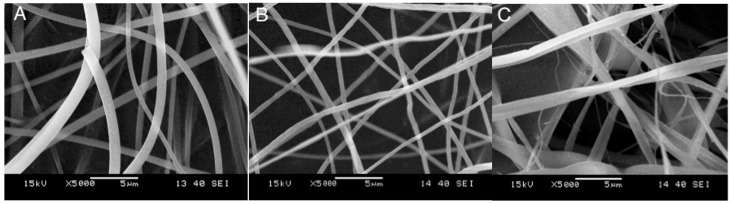
SEM micrographs recorded on the electrospun fibers prepared from the three carrier polymers: (**A**) PVP, (**B**) PVA, and (**C**) HPMC.

**Figure 4 molecules-28-04188-f004:**
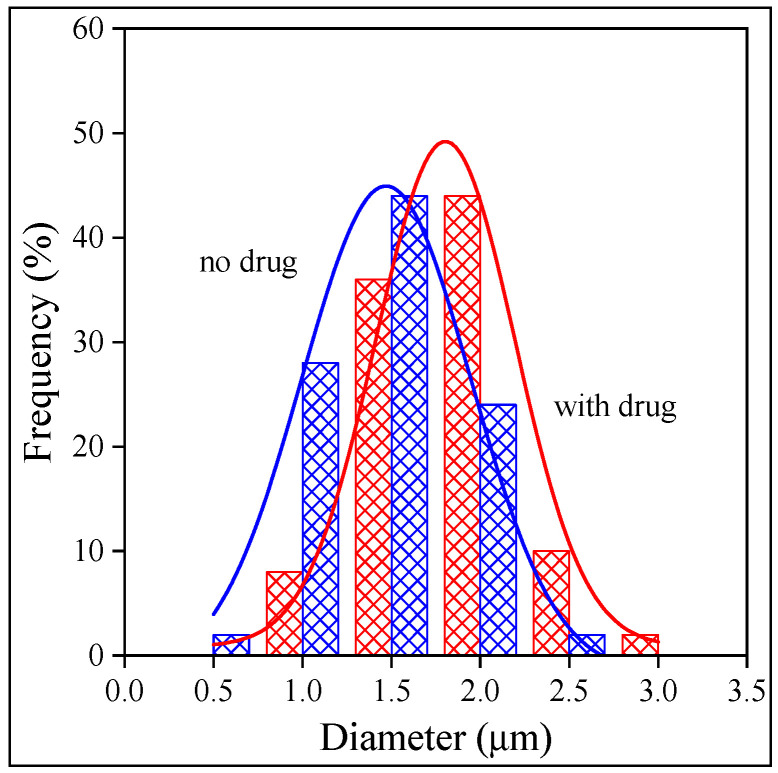
Size distribution of HPMC electrospun fibers with and without the drug (Levo).

**Figure 5 molecules-28-04188-f005:**
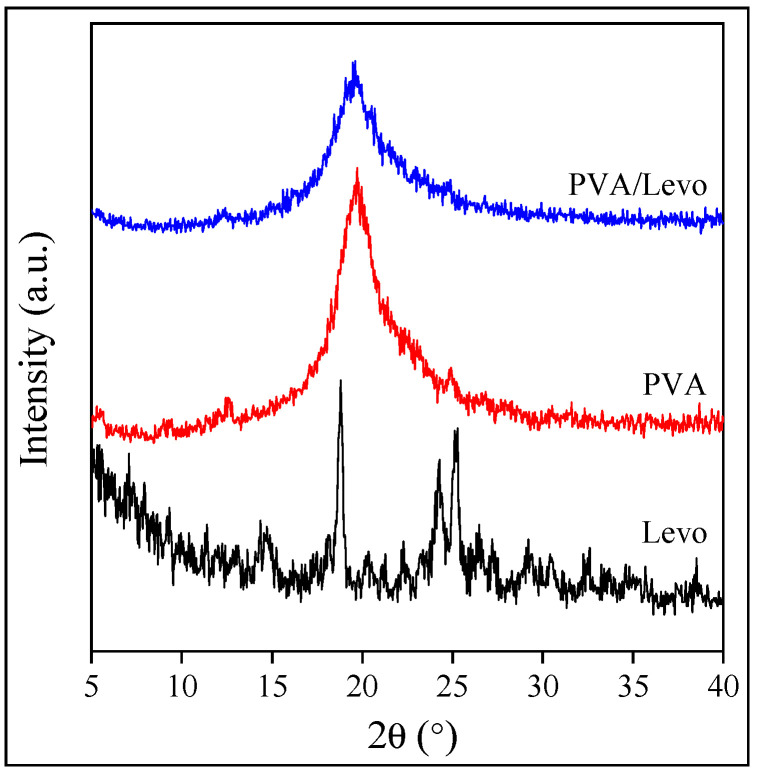
Structure of PVA electrospun fibers containing Levo. Comparison of XRD patterns.

**Figure 6 molecules-28-04188-f006:**
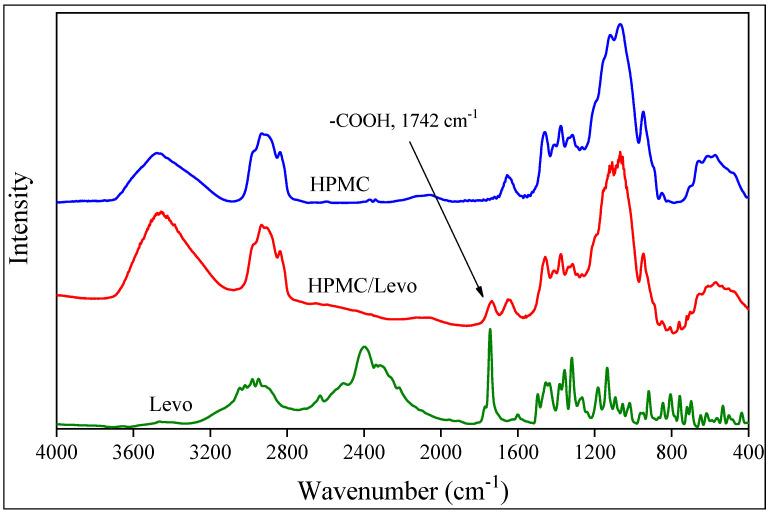
Comparison of the FTIR spectra of HPMC for Levo and fibers containing the drug; proof of encapsulation.

**Figure 7 molecules-28-04188-f007:**
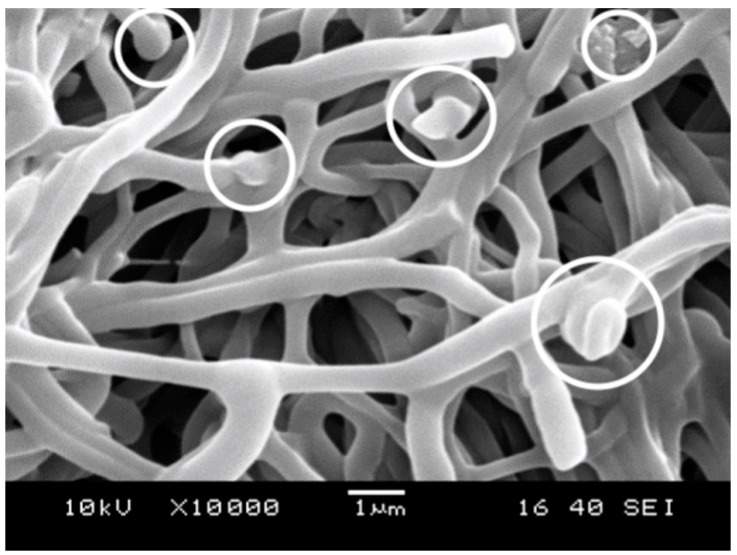
SEM micrograph recorded on PVA fibers showing the encapsulation of the drug as separate phase.

**Figure 8 molecules-28-04188-f008:**
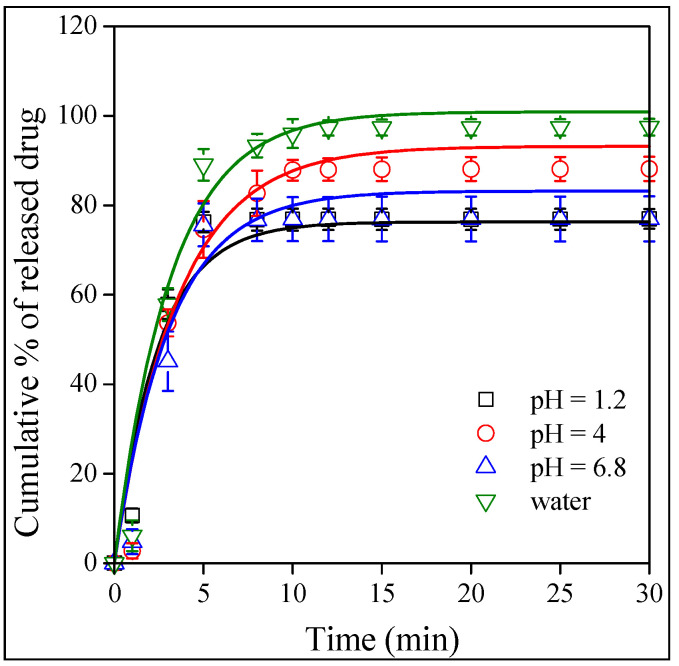
Time dependence of the dissolution of neat levocetirizine in water and in buffers with various pH values: (☐) 1.2, (◯) 4.0, and (△) 6.8 (▽) distilled water.

**Figure 9 molecules-28-04188-f009:**
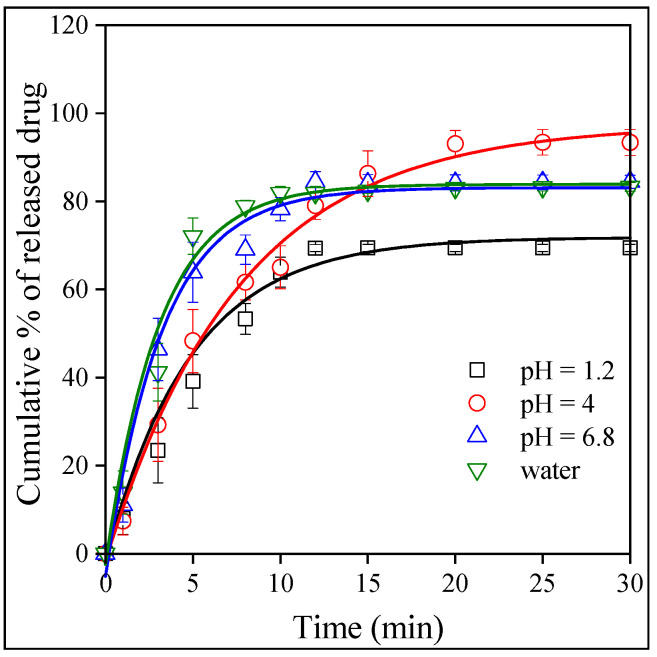
Effect of pH on the release of Levo from HPMC fibers. Symbols: (☐) 1.2, (◯) 4.0 and (△) 6.8 (▽) pH distilled water.

**Figure 10 molecules-28-04188-f010:**
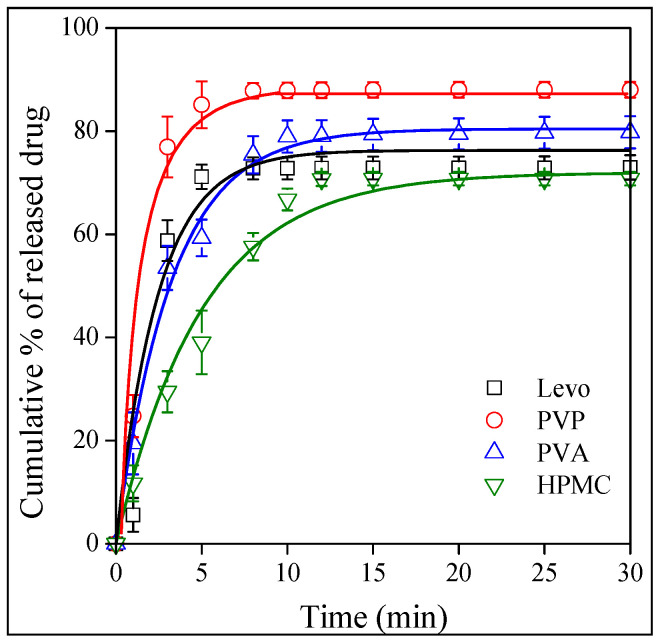
Comparison of the dissolution of Levo and the release characteristics fibers prepared from the three polymers at pH of 1.2. Symbols: (☐) neat Levo, (◯) PVP, (△) PVA and (▽) HPMC.

**Figure 11 molecules-28-04188-f011:**
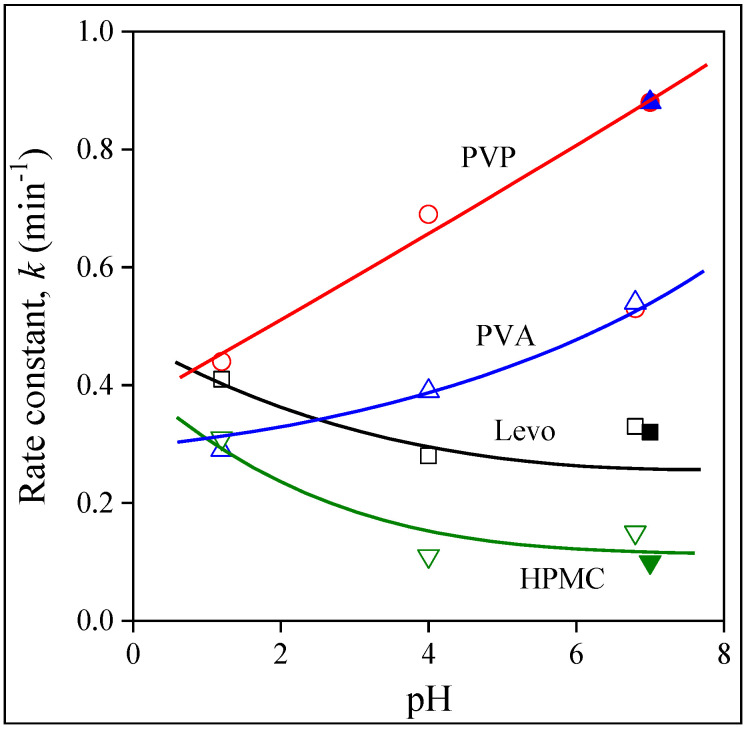
Effect of polymer type and pH on the overall rate of dissolution and drug release. Symbols: (☐) neat Levo, (◯) PVP, (△) PVA and (▽) HPMC. Filled symbols indicate values determined in distilled water.

**Table 1 molecules-28-04188-t001:** Solubility parameters of the components and the solubility of the drug in them [45].

Component	Solubility Parameter, δ (MPa^1/2^)	Solubility (wt%)
PVA	33.0	0.78
PVP	26.0	2.00
HPMC	28.6	0.77
Levocetirizine	22.9	n.a.
Dichloromethane	20.2	6.7
Ethanol	25.7	50
Water	47.9	96

**Table 2 molecules-28-04188-t002:** Optimized technological conditions used for the electrospinning of fibers prepared from the three water-soluble polymers of this study.

Polymer	Concentration (wt%)	Voltage (kV)	Feeding Rate (μL/s)	Collector Distance (mm)
PVP	40	18	0.5	100
PVA	15	15	0.2	140
HPMC	10	18	0.5	125

**Table 3 molecules-28-04188-t003:** Diameter of the fibers spun from the polymers used in the study and the effect of the incorporation of the drug into them.

Polymer	Average Diameter (µm)	
	No Drug	With Drug
HPMC	2.7 ± 1.5	2.9 ± 1.9
PVP	3.1 ± 0.1	4.2 ± 0.2
PVA	1.5 ± 0.1	1.9 ± 0.2

## Data Availability

The materials and datasets generated and analyzed during the current study are publicly available from the corresponding author on reasonable request.

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
