# Peer review of "Levocetirizine-Loaded Electrospun Fibers from Water-Soluble Polymers: Encapsulation and Drug Release"

_molecules, 2023, doi:10.3390/molecules28104188_

Round 1

Reviewer 1 Report

Li et al. designed and developed Levocetirizine-loaded electrospun nanofibers using three different water-soluble polymers and further investigated their drug release behavior. Some major concerns and issues are suggested to be addressed before publication.

1. The title is too broad and a little bit misleading. It is suggested to be rewritten in a more specific manner.

2. Abstract: The authors claimed that Amorphous drug was homogeneously distributed within fibers. How did the authors prove this conclusion?

3. The authors utilized three different polymers to load the drug. Did the authors explain which one is best?

4. Please state the reasons why three different water-soluble polymers were chosen in this study. What are the merits and demerits of water-soluble polymers compared with the un-soluble ones?

5. The merits of electrospinning technique should be further outlined, and some recent works about the innovative electrospinning like 10.3390/nano13071150 and 10.1016/j.eurpolymj.2023.111863 are suggested to be discussed.

6. In the Materials section, some important information are missing. For example, how about the molecular weight of each polymer and purity degree of each chemical?

7. How did the authors select the solution concentration? Did they conduct any pre-studies? In addition, the electrospinning parameters should be given.

8. Figure 3: The mean fiber diameter and fiber diameter distribution of each electrospinning sample are suggested to be added.  

9. Did the fiber diameter affect the drug release?

10. Figure 8-10: How many replicates were employed for each sample? The data should be presented in the form of mean ± standard deviation (SD).

11. The grammar and writing should be improved in the whole manuscript.

Moderate editing of English language is required.

Reviewer 2 Report

This paper describes synthesis of three electrospun fibers loaded with levocetirizine for DDS devices along with their characterization and release. The results may be correct. Unfortunately, comparative discussions on three fibers, crystalline levocetirizine, and results in references 37–39 are insufficient, which needs to include advantages of the present fibers over other methods. This referee recommends publication of this work in Molecules provided that suitable revisions are made.

Author Response

We appreciate the comments of the Referees as well their suggestions to improve the quality of our paper. We did our best to take all of them into account during revision and modify the paper accordingly. The modifications carried out and the answers to the questions are listed below in detail. Questions are printed by normal letters, while answers are written in italic.

Reviewer #2

     This paper describes synthesis of three electrospun fibers loaded with levocetirizine for DDS devices along with their characterization and release. The results may be correct. Unfortunately, comparative discussions on three fibers, crystalline levocetirizine, and results in references 37–39 are insufficient, which needs to include advantages of the present fibers over other methods. This referee recommends publication of this work in Molecules provided that suitable revisions are made.

The Reviewer requires the revision of the results based on the case that comparative discussions on three fibers, crystalline levocetirizine, and results in references 37–39 are insufficient. As we s in the last paragraph of the introduction results published in the open literature indicate that the studies undertaken with levocetirizine up to now had various goals. Some wanted to increase release rate, while others aimed at prolonged release, or simply intended to mask the bitter taste of the drug. Considering these, sometimes contradictory aims and results, the goal of our study was to prepare formulations from Levo by incorporating it into electrospun water soluble polymer fibers. Data available in the existing literature are sometimes confusing and make it impossible to compare the results of different studies because of the different materials, characterization methods and conditions.

Reviewer 3 Report

The authors reported on the preparation of electrospun fibers containing Levocetirizine, a BCS III drug, from three different water-soluble polymers - HPMC, PVP, and PVA. They also optimized the fiber spinning technology for each polymer to achieve the desired drug release.

However, the design of the experiments and quality of the presentation are not good enough. I only recommend the publication upon the following conditions are well addressed.

[1] Could this encapsulation method also be applied to other drugs especially BSC III drugs besides Levocetirizine? The generalizability of this method to other drugs are better further explored.

[2] The authors did not report on the long-term stability of the electrospun fibers containing Levocetirizine. The stability of these fibers over time is crucial for their practical application as drug delivery systems.

[3] The authors optimized fiber spinning technology for each polymer separately and thoroughly characterized the morphology of the fibers. They did not investigate other factors that could affect drug release kinetics, such as temperature and humidity.

[4] The English writing of this manuscript is not good enough. Please find native English speakers to check it.

[5] Some sentences are written very generally not specific enough, e.g., (1) “These interactions, depend on the chemical and physical characteristics of the components. Levocetirizine contains several rings and an acidic group, the latter being able to form strong interactions with some of the carrier polymers used in this study” from line 183 to line 186; (2) “Small molecular weight polar compounds usually dissolve to a very small extent in polymers with smaller polarity” from line 211 to line 212.

[6] In Figure 1b and Figure 5, the y-axis should be correct into “” or “Diffraction angles ” but not angle of reflection.

[7] Don’t mix up the use of “anymore” and “any more” in the manuscript.

[8] In Table 1, it is not clear solubility parameters and solubility representing for, particularly for Dichloromethane, Ethanol and Water.

[9] Please rewrite the abstract which is almost identical to the conclusion.

[4] The English writing of this manuscript is not good enough. Please find native English speakers to check it.

Author Response

        We appreciate the comments of the Referees as well their suggestions to improve the quality of our paper. We did our best to take all of them into account during revision and modify the paper accordingly. The modifications carried out and the answers to the questions are listed below in detail. Questions are printed by normal letters, while answers are written in italic.

Reviewer #3

The authors reported on the preparation of electrospun fibers containing Levocetirizine, a BCS III drug, from three different water-soluble polymers - HPMC, PVP, and PVA. They also optimized the fiber spinning technology for each polymer to achieve the desired drug release.

However, the design of the experiments and quality of the presentation are not good enough. I only recommend the publication upon the following conditions are well addressed.

  1. Could this encapsulation method also be applied to other drugs especially BSC III drugs besides Levocetirizine? The generalizability of this method to other drugs are better further explored.

We thank the Referee for her or his question. The encapsulation method can be used for other drugs and we successfully encapsulated drugs (e.g. valsartan, metronidazole, pregabalin etc) both in water-soluble and insoluble polymers.

  1. The authors did not report on the long-term stability of the electrospun fibers containing Levocetirizine. The stability of these fibers over time is crucial for their practical application as drug delivery systems.

We thank the Referee for her or his question. We appreciate the comment of the Reviewer and we agree with him or her completely that the stability of fibers could be crucial for their application but until now we did not investigate this issue.

  1. The authors optimized fiber spinning technology for each polymer separately and thoroughly characterized the morphology of the fibers. They did not investigate other factors that could affect drug release kinetics, such as temperature and humidity.

We thank the Referee for her or his comment regarding our thorough characterization and we agree with him or her completely that temperature of the experiments affects the kinetics of drug release. On the other hand, it is important to note that the investigation of the effect of temperature on kinetics of drug release are out of scope of the work.

  1. The English writing of this manuscript is not good enough. Please find native English speakers to check it.

We thank the Referee for her or his comment regarding the English writing. On the other hand, the referee did not specify how can we improve the quality of English Language. Accordingly, the language was checked thoroughly throughout the paper to clarify the meaning of the sentences and thus avoid misunderstanding.

  1. Some sentences are written very generally not specific enough, e.g., (1) “These interactions, depend on the chemical and physical characteristics of the components. Levocetirizine contains several rings and an acidic group, the latter being able to form strong interactions with some of the carrier polymers used in this study” from line 183 to line 186; (2) “Small molecular weight polar compounds usually dissolve to a very small extent in polymers with smaller polarity” from line 211 to line 212.

We thank the Referee for her or his comment, the text has been modified accordingly

These interactions, depend on the chemical and physical characteristics of the components. Levocetirizine contains several rings and an acidic group, the latter being able to form strong intermolecular hydrogen bonding with some of the carrier polymers used in this study”

Small molecular weight polar compounds usually dissolve to a very small extent in polymers with smaller polarity. Billingham et al. [44] for example reported that dissolution of various stabilizers in polypropylene and polyethylene is cannot be more than 0.1 or 0.2 wt%.”

  1. In Figure 1b and Figure 5, the y-axis should be correct into “” or “Diffraction angles ” but not angle of reflection.

We thank the Referee for her or his comment, the figures have been modified accordingly.

  1. Don’t mix up the use of “anymore” and “any more” in the manuscript.

We thank the Referee for her or his comment, the text has been modified accordingly

When the solubility limit is reached, the drug precipitates and forms a separate phase. Accordingly, the intensity of absorption does not increase anymore, or just proportionally to the surface of the particles dispersed in the polymer.”

  1. In Table 1, it is not clear solubility parameters and solubility representing for, particularly for Dichloromethane, Ethanol and Water.

Component

Solubility parameter, δ

(MPa1/2)

Solubility

(wt%)

PVA

33.0

0.78

PVP

26.0

2.00

HPMC

28.6

0.77

Levocetirizine

22.9

n.a.

Dichloromethane

20.2

6.7

Ethanol

25.7

50

Water

47.9

96

Table 2 shows the solubility of the drug in the three polymers (rows 1-3) and the solubility of the drug in the solvents used in this experiment (row 4-7). Solubility parameters estimated by Fedor’s method for the main components used in the technology, while solubility of the drug in the components was measured by us.

  1. Please rewrite the abstract which is almost identical to the conclusion.

According to the request of the Reviewer, the title of the paper was modified.

Round 2

Reviewer 1 Report

The reviewer's comments have been addressed.

Reviewer 3 Report

It could be accepeted in the present form.

Moderate editing of English language